# Blue Economy and Blue Activities: Opportunities, Challenges, and Recommendations for The Bahamas

Brandon J. Bethel [1] , Yana Buravleva [2] and Decai Tang [3,4,*]

1    School of Marine Sciences, Nanjing University of Information Science & Technology, Nanjing 210044, China; 20195109101@nuist.edu.cn
2    School of Business, Nanjing University of Information Science & Technology, Nanjing 210044, China; 20195225101@nuist.edu.cn
3    China Institute of Manufacturing Development, Nanjing University of Information Science & Technology, Nanjing 210044, China
4    School of Management Science and Engineering, Nanjing University of Information Science & Technology, Nanjing 210044, China
*    Correspondence: 002032@nuist.edu.cn

**Abstract:** Following the global shutdown of tourism at the onset of the COVID-19 pandemic, small island developing states such as The Bahamas had their economies immobilized due to their heavy dependence on the industry. Beyond economic recovery in a post COVID-19 paradigm, the blue economy, blue growth, and associated activities offer pathways for a more resilient economy and is well-suited for The Bahamas. This paper suggests conduits for economic development using a traditional strength, coastal and marine tourism, in conjunction with the emerging fields of ocean renewable energy, offshore aquaculture, marine biotechnology, and bioprospecting. The interlinkages between each activity are discussed. Knowledge gaps in offshore aquaculture, ocean renewable energy, marine biotechnology, and marine environment monitoring are identified. In each sector case, strategic and tactical decision-making can be achieved through the exploitation of ocean numerical modeling and observations, and consequently should be invested in and developed alongside the requisite computational resources. Blue growth is encouraged, but instances of blue injustice are also highlighted. Crucially, pursuing blue economy activities should be given top national priority for economic recovery and prosperity.

**Keywords:** blue economy; blue growth; small island developing states; The Bahamas; ocean numerical modeling; ocean observations



## 1. Introduction

When seen at a distance of approximately six billion kilometers by the Voyager 1 space probe on February 14, 1990, the Earth was described by Carl Sagan as a "pale blue dot", where, excluding the planet's insignificant size against the cold backdrop of space, only the colors of the ocean were captured by the now famous photograph. Twenty-five years later, U.S. astronaut Scott Kelly, in 2015, named The Bahamas "the most beautiful place from space", with his photos sharing one crucial element with the Voyager 1 image: the colors of the ocean. By the end of 2019, The Bahamas had experienced record levels of foreign air and sea arrivals that totaled over 6.6 million visitors, up from 6.1 million in 2017. By early 2020, however, the tourist industry crashed in the wake of the COVID-19 pandemic where the country saw only 422,000 stopover visitors, down from a little over 1.8 million in the previous year. This represents a colossal decrease of 63.9%. While the Government has unveiled a host of measures to restart the economy, a recent USD 200 million loan from the Inter-American Development Bank (IDB) offers another avenue for recovery: the blue economy. The blue economy can broadly be defined as "the sustainable use of ocean resources for economic growth, improved livelihoods and jobs, and ocean

ecosystem health, and covers interlinked established and emerging activities such as fisheries, tourism, maritime transport, offshore renewable energy, aquaculture, ocean mining, marine biotechnology, and bioprospecting" [1].

The ocean economy, the series of economic activities linked to the ocean, inclusive of assets, goods, and services of marine ecosystems, has been valued at approximately USD 24 trillion [2] and is predicted to grow faster than the global economy from 2010 to 2030 [3,4]. This is especially true for coastal and island states [1,5], of which The Bahamas, as a small island developing state (SIDS), is one. Blue recovery and blue growth, therefore, are strategies to restart and revitalize economies through those and other activities. It is argued that following successful examples of economies being supported by the blue economy [6–9], The Bahamas and other SIDS can initiate transformations of their own to revolve around their oceans to not only recover from catastrophic economic losses experienced due to the shutdown of global tourism but accelerate economic growth to beyond their pre-COVID-19 levels. Additionally, a variety of United Nations Sustainability Goals (SDGs) can be addressed through engaging in blue economy activities. This idea stimulated this study into how The Bahamas can get on a path of national recovery through a comprehensive review of global literature. Challenges and barriers are discussed, and possible solutions provided. The rest of this paper is structured as follows: Section 2, in sequence, discusses coastal and marine tourism, fisheries and offshore aquaculture, marine bioprospecting and biotechnology, and ocean renewable energy. Section 3 presents challenges to blue economy activities inclusive of instances of blue justice and deficiencies in ocean observations. Section 4 summarizes the findings, provides recommendations, and concludes.

## 2. Overview and Opportunities

### 2.1. Coastal and Marine Tourism

Globally, tourism and travel combined account for nearly ten percent of GDP and brings in an annual review of approximately USD 7.6 trillion, and thus is an important part of the world economy. For The Bahamas, the tourism economy and tourist sectors account for 45% and ~25% of GDP, respectively, both dwarfing the regional averages of 30% and 11% [10,11]. Following the onset of the COVID-19 pandemic, estimates of job losses for the Caribbean ranged from 1.4 million to 2 million, and economic loss in the tourism sector was measured between USD 27 billion to USD 44 billion with recovery expected to take anywhere from ten months to two years, and may be longer for smaller economies reliant on tourist arrivals from a few developed economies [12]. Approximately 89% of The Bahamas' tourist arrivals are from the United States and Canada alone. The industry, as with the nation itself, is at risk from climate change [13–15]. Although several products have been proposed by the Ministry of Tourism to restart the industry, the Blue Economy can also be relied upon to initiate a sustainable and inclusive recovery. Indeed, before COVID-19, ocean tourism was valued at USD 390 billion and constitutes a significant portion of the GDP of many nations, and alone, could account for 26% of ocean-based economic activity by 2030 [16].

To engender blue recovery through coastal tourism, incorporating ecotourism into design strategies has been shown to be an effective route and has the added benefit of promoting environmental conservation and minimizing environmental impact [17,18]. The maximization of carbon sequestration, wild fish stock recovery, improved water quality, and reef/ecosystem health is dependent on the expansion of protections for wetlands, mangroves, and seagrass fields. The economic value of these habitats is immense. For example, sharks and rays that are born in these environments have been documented to generate USD 113.8 million annually for The Bahamas in the world's largest shark diving industry [19]. As a global leader in ecotourism showcasing vacations at national parks that are almost entirely composed of these coastal features, The Bahamas already has a strong incentive to minimize their decay and should invest significant resources in fortifying and restoring coastlines, beaches, coral reefs, and marine biodiversity. For

example, Arkema et al. discovered that visitation to natural and built environments were well correlated with the count of visitors from entrance surveys for islands and parks and made it possible to predict visitor days to marine protected areas (MPAs) [20]. There, tourists preferred accessible natural landscapes, such as reefs near lodges, and they spent 125,000 visitor nights and more than USD 45 million in the most highly visited district, over five times the least visited district. Coupling sustainable marine and domestic tourism is another pathway towards revitalizing the industry. In a recent survey, 84%, 69%, 68%, and 63% of respondents from China, France, Spain, and the United States, respectively, have suggested that in response to the COVID-19 pandemic, they would make their next leisure trip a domestic one [21]. Cultivating domestic tourists can have the effect of remedying the lack of interisland linkages, both economic and transport, that have been cited as factors in restricting economic growth [22–24]. Caution should be exercised as pre-existing sustainability issues in the marine tourism sectors could potentially be exacerbated if blue growth is allowed unrestrained by sustainability considerations.

*2.2. Fisheries and Offshore Aquaculture*

Bahamian fisheries account for 2% of national GDP and, though dwarfed by tourism at more than 45%, remains economically, culturally, and nutritionally important [25,26]. The blue economy and its principles can also be used to foster greater economic growth through triggering sustainable growth in this industry. Traditionally important species as Queen Conch (*Lobatus gigas*), Caribbean spiny lobster (*Panulirus argus*), and Nassau grouper (*Epinephelus striatus*) generate over USD 1.4 billion in commercial landings for The Bahamas over the past two decades [26], though these species are currently at risk due to overexploitation. Bonefish (*Albula* spp.) are the focus of Bahamian recreational flats fishery and generates USD 140 million per year, with the bulk going to the Family Islands, but due to increasing sea surface temperatures, they may become more vulnerable. In each case, MPAs have and should be designed to account for genetic structure throughout the region, foraging habitats, migration routes, spawning aggregations, larval dispersal routes, and particularly for bonefish, thermal refuges [27–30].

For additional protection, the industry should be directed on a more sustainable path and be made reliant on mapping small-scale fisheries (SSF), as these comprise the largest global group of ocean users [31]. If ocean governance is not enacted to better account for the social dimensions of fisheries, this group may fall victim to exclusion to the dialogue between international environmental and economic factors that determine strategies for the future of the ocean. This is important because, as Cohen et al. [32] argues, SSFs may be subtly or overtly pressured for geographic, political, and economic space by larger commercial/environmental conservation entities, thus jeopardizing both food and economic security. SSFs may be excluded from decision-making processes during marine spatial planning and MPA activities. Cohen et al. [32] and Bennet [33] argue that for development to be both inclusive and sustainable, enhanced usage of collaborative and integrative elements, and data on multiple dimensions of the trade-offs being negotiated in MSP activities would be required. Additionally, protected areas should also be expanded beyond the 20% currently mandated by the Government as they are vital to sustainable coastal and marine tourism and the ecological spillover effects they induce [34]. A transformation of traditional fisheries modes into others may be required for sustainability. Noting that definitions vary by region, marine ranching in the Chinese context is the sustainable fishery mode that from an ecological perspective "facilitates the breeding and conservation of fishery resource and marine eco-environment improvement using various measures such as artificial reefs, stock enhancement, and releasing, to construct or restore breeding, growth, forage, and shelter habitats for marine organisms" [35]. Marine ranching naturally has strong implications for the fishing industry, conservation efforts, and is at an intersection with aquaculture.

Complementary to fisheries through conservation [36,37], offshore aquaculture (or marine aquaculture) has emerged as a crucial component in the blue economy [38]. For

The Bahamas, a recent study by Thomas et al. [39] demonstrated that a large potential for cobia (Rachycentron canadum) aquaculture production exists and is due primarily to its relatively large exclusive economic zone and extensive shelf. The authors note, however, that the majority of cobia farms are not profitable due to cooler average sea surface temperatures retarding cobia growth rates. This result was observed in a previous investigation conducted in The Bahamas and Puerto Rico by Benetti et al. [40]. Parrotfish (Scarinae) have been suggested by Sherman et al. [41] as an emerging species for fisheries, but due to their importance in maintaining reef health, more research is required for assessing the viability of this plan and should be balanced against the species' ecological value, and the services the reefs themselves provide to The Bahamas. Targeting this species for marine ranching may help to develop the industry in an ecologically responsible way. Other species, commercially, culturally, or ecologically important, endangered or otherwise, can also be targeted, in full consultation with, and cooperation of, SSFs.

Several factors inhibit local aquaculture development. These include, but are not limited to, the physical (low tidal range that can flush ponds or cages, and high evaporation rates), chemical (excluding Andros and the Abacos, scarce freshwater resources limit culture operations to brackish or marine species, and minute nitrogen and phosphorous concentrations), or biological (a lack of plankton, due to small nitrogen and phosphorous concentrations), but also include the technological and societal. Although aquaculture research and development in The Bahamas is supported by several entities (e.g., Tropic Seafood, The Island School, and The Bahamas Agriculture and Marine Science Institute, etc.), higher levels of research are required to generate the requisite biological information for the large-scale commercial operations such as those practiced by Tropic Seafood. A mechanism by which private entities are incentivized to share information and skills generated with potential competitors should be put in place, with skilled and experienced personnel at all levels of aquaculture science and management culture. Partnerships with international organizations, universities, and research facilities should be encouraged. For The Bahamas to evolve out of the experimental stage, political will and commitment and viable primary resource assets to support commercial culture are progress prerequisites [42]. At the time of that report, there were no aquaculture policies and plans available to incorporate into larger national development plans and, as such, there was inadequate support for development. With the passing of the Fisheries Bill, 2020 that repealed the Fisheries Resources (Jurisdiction and Conservation) [43] Act, 1977, specific language for aquaculture management and development has been included and should bode well for continued progress [44].

Although promotion of marine aquaculture will undoubtedly lead to economic diversification, growth, and can increase food security, expectations must not exceed actual capacity and should not be allowed to marginalize SSFs [45]. This leads to the requirement that a holistic assessment of marine aquaculture's potential in The Bahamas to support its blue economy aspirations be conducted that takes into consideration other users of maritime space and the potential to exclude SSFs. This assessment should cover biological, technical, and economic constraints to test the estimation of Costello et al. [46] that suggested that the oceans can provide up to six times the amount of food it produces today with better management and innovation.

### 2.3. Marine Bioprospecting and Biotechnology

As the ocean is the largest reservoir of the world's animal and plant species and subjects these species to a wide range of thermal, pressure, and nutrient pressures with extensive photic and aphotic zones, evolution has produced a dazzling array of unique genes, molecules, and compounds that are waiting for their systematic discovery through marine bioprospecting [47], and application through marine biotechnology. Marine biotechnology encompasses fields such as marine drug discovery [48], antifouling solutions [49], food production and processing [50], cosmeceuticals [51], and energy [41,52], amongst other sectors [53]. With the recent tabling of The Biological Resources and Traditional

Knowledge Protection and Sustainable Use Act, 2020 (hereinafter, Biological Act), local enterprises can join their regional and international counterparts by establishing centers for marine bioprospecting, biotechnology, and biomedicine, amongst other activities [54]. This allows firms and research centers to take advantage of the country's genetic resources for economic, drug, food, and energy development, thus leading to increasing levels of economic diversification. As the country has a total reef area of 3580 km$^2$ and hosts the world's third largest barrier reef system (the Andros Barrier Reef of area ~270 km$^2$), the nation is a significant marine genetic repository. Hargreaves-Allen presented an economic valuation of the natural resources of Andros Island and though noted that coral reef organisms are a reservoir for biomedically important substances, a specific valuation from a marine biotechnology and prospecting perspective was not given [55]. This is a significant blind spot that retards industry development. Extensive phylogeographic studies would be required not only in marine bioprospecting activities but would also aid in devising marine management schemes [56] and should consequently be made a national priority.

Excluding the Biological Act, regulatory frameworks to manage the commercial exploitation of biodiversity to the best of the author's knowledge do not currently exist, thus necessitating examples from other jurisdictions to be drawn upon. In Indonesia, the world's largest archipelagic state and a major player in marine bioprospecting, Siswandi, [57] argued that marine bioprospecting frameworks should regulate access to, and the sharing of benefits from, marine genetic resources and should be developed in the context of sustainable development and the blue economy. Based on the Indonesian experience, specific rules and regulations should be formulated for a Bahamian context to govern access to genetic resources, marine scientific research, and patent law. The Marine Microbial Biodiversity, Bioinformatics and Biotechnology (Micro B3) project, supported by the European Union from 2012–2015, was geared towards developing intellectual property agreements for "the protection and sustainable use of pre-competitive microbial genetic resources and their exploitation in high potential commercial applications" [58], in addition to its primary purpose of developing bioinformatic approaches and a legal framework to make large-scale viral, bacteria, archaeal and protists genomes, and metagenomics accessible for marine ecosystems biology and to define new targets for biotechnological applications [59] and, as such, can be undertaken locally to develop the industry. Similarly, data that meet the principles of findability, accessibility, interoperability, and reusability (FAIR) can offer additional benefits such as the ability of machines to automatically find and use the data, in addition to supporting reuse by individuals [60,61]. Emphasis on the commercial applications of biodiversity should not come at the expense of environmental responsibility. Combating biopiracy may initially be a problem but though it may be popular to affirm sovereign right over natural resources, policies to ensure regional and global accessibility to marine biodiversity hotspots for the marine bioprospecting of pharmacologically significant products can lead to greater benefits beyond the financial [62,63].

### 2.4. Ocean Renewable Energy

Energy, and its affordable access, are amongst the most important indicators of economic health and is a United Nations Sustainability Development Goal (SDG7), but is also often unstable, and are sources of pollution for SIDS. High energy costs, and its instability, are major impediments to efficient production and economic growth in The Bahamas [22]. To undergird the blue economy and all other economic sectors, access to carbon neutral/negative electricity should be made a national priority as continued reliance on hydrocarbons and fossil fuels is not a viable long-term strategy, considering primarily global market volatility and large import and domestic transportation costs, not to mention the $CO_2$ emissions they invariably produce. Renewable energy systems offer a chance to reduce electricity production costs and can attract aid from international development partners to further reduce costs associated with these activities. For the Caribbean, IDB and International Bank for Reconstruction and Development are amongst the largest contributors to energy development aid energy SIDS from 2012–2016, with this aid shifting

toward cleaner (and in particular, solar) energy sources [64]. As a case in point, renewable energy was designated as the optimal alternative to increase energy security in Cape Verde, though there may be trade-offs between risk mitigation and poverty alleviation benefits of different renewable technology investments [65].

Ocean renewable energy (ORE) conventionally includes waves, kinetic (tides and currents), ocean thermal energy conversion, and salinity gradients, but can also include offshore wind and solar given their installation in marine environments. ORE is a major component of the blue economy but, excluding a photovoltaic microgrid installed on Ragged Island that has a capacity of 402 kW and other small-scale solar installations, renewable energy in general is limited in The Bahamas, accounting for <0.1% of total electricity generation. In the Bahamas Energy Snapshot prepared by the National Renewable Energy Laboratory (NREL), the status and potential of various sources of renewable energy was collated [66] and is shown graphically in Figure 1. There, the potential of biomass, solar, and wind energy were measured at 1, 60, and 200 MW, respectively, though no details as to how the potentials were calculated and, thus, no independent assessment can be conducted. Crucially in the NREL estimation, wind and solar resources offered the greatest potential for renewable energy development in The Bahamas, but it should be noted that the potential of the ocean was not quantified. Given NREL's defined ORE potential status as "unknown", this represents a significant blind spot and a major hurdle in assessing and exploiting its actual renewable energy resources.

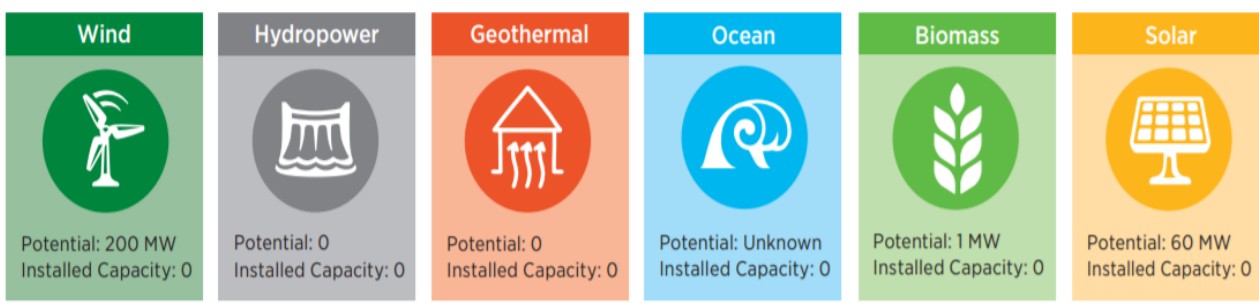

**Figure 1.** Renewable energy status and potential (adapted from NREL, 2015 [66]).

A review of the literature finds that wind and wave energy assessments have been carried out for the Caribbean, but none, to date, have focused specifically on The Bahamas. Instead, Table 1 gives the energy resources for The Bahamas extracted from the most recent regional or global energy assessment. Two sources of ORE stand out for their current lack of certainty. For tidal current power, due to the region's microtidal climate, with a tidal range of ~0.2 m [67], extracting energy from tides is perhaps commercially unviable [68], though new technologies are being developed for slower tidal current velocities which may change this assessment [69].

**Table 1.** Energy resource densities for The Bahamas collated from the literature.

| Resource Type | Unit | Study Range (years) | Maximum | Minimum | Citation |
|---|---|---|---|---|---|
| Wind | W/m$^2$ | 31 | 300 | 200 | Chadee and Clarke [70] |
| Waves | kW/m | 30 | 15 | 2 | Guillou and Chapalain [71] |
| Thermal * | kW/m$^2$ | 1000 | 250 | 200 | Rajagopalan and Nihous [72] |
| Salinity | Wm$^3$/s | 114 | 0 | 0 | Alvarez-Silva et al. [73] |
| Tidal ** | m/s$^2$ | - | 0 | 0 | Roberts et al. [68] |
| Currents | kW/m$^2$ | 2 | 2 | 0.5 | Bane et al. [74] |

* Indicates the power density with no feedback from OTEC plants. ** Calculated for a barrage or lagoon.

To estimate salinity gradient energy (SGE), salinity differentials must exist and, as such, river mouths are essentially the only locations where this form of energy is available, and this is reflected by the literature [73,75]. The Bahamas does not possess any rivers

and, as such, its potential to generate energy through pressure-retarded osmosis, reverse electrodialysis, capacitive mixing, or capacitive reverse electrodialysis at the ocean–river interface is exactly zero. However, large volumes of freshwater exist in lenses in Andros Island and through a special application of those available technologies, SGE may not actually be zero, requiring research into this possibility of SGE generation.

To uncouple its economy from the volatile fossil fuel industry, reduce its imports of hydrocarbons and greenhouse gas emissions (thereby minimizing its already small contribution to anthropogenic climate change), The Bahamas should incentivize ORE and should begin with local high-resolution energy assessments alongside considering less energy-demanding modes of societal operation such as circular economies [76] or engaging in smart mobility [77]. ORE engenders short- and long-term job creation, improves local air quality [12], and provides secondary benefits such as increasing resiliency to future global shocks, water security through desalination [78], and energy storage through the production of energy carriers such as hydrogen [79]. Directly, however, if an example is taken from Yang et al. [80], where they estimated the ocean current potential for the Gulf Stream system that flows through the Florida Strait between the United States and The Bahamas, if assuming a power efficiency of 30% from energy removal from flow to electrical power, turbines could yield a peak of approximately 13 GW. If, in a perfect world, power is shared between The Bahamas and the United States equally, the 6.5 GW dwarves both the Bahamas Power and Light and Grand Bahama Power Company combined installed capacity of 536 MW [81], and the combined peak demand of [64]. This corresponds to ~22.5 TWh/y (half the value of 45 TWh/y from Yang et al. [82]), which greatly exceeds 1930 GWh for the country.

## 3. Challenges to Blue Growth

### 3.1. Blue Injustice

Similar to all jurisdictions, The Bahamas is far from achieving perfect governance, and blue growth may generate or exacerbate pre-existing injustices. Bennet et al. outlined ten social injustices: (1) dispossession, displacement, and ocean grabbing; (2) environmental justice concerns from pollution and waste; (3) environmental degradation and reduction of ecosystem services; (4) livelihood impacts for small-scale fishers; (5) lost access to marine resources needed for food security and wellbeing; (6) inequitable distribution of economic benefits; (7) social and cultural impacts; (8) marginalization of women; (9) human and Indigenous rights abuses; and (10) exclusion from governance [5]. The Bahamas should consider these and identify other potential blue injustices that may arise to its own specific implementation of the blue economy, particularly as injustices 3–5 were recently brought to the forefront by oil drilling activities conducted by the Bahamas Petroleum Company in Southern Bain, Cooper, Eneas, and Donald licenses (Figure 2).

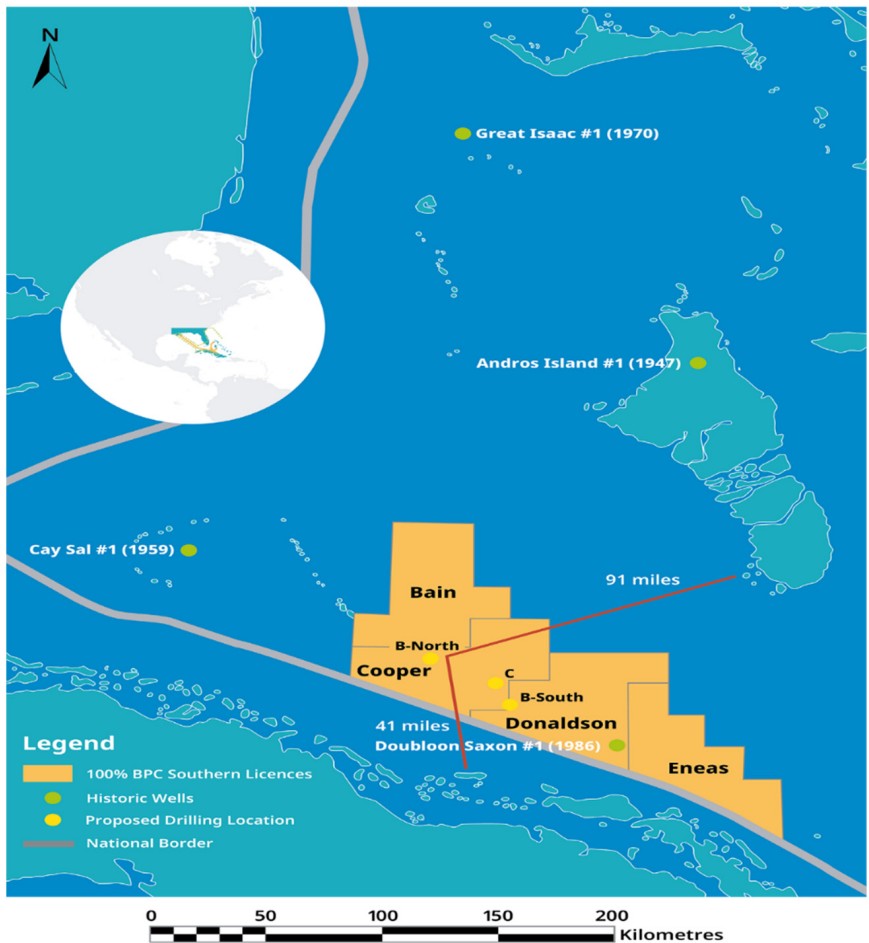

**Figure 2.** Historic wells, proposed drilling locations and Southern Licences (adapted from BPC [83]).

Although BPC had released an Environmental Impact Assessment [84] before drilling commenced, as required by law, opposition from fishermen and local and international environmental groups was intense [85]. Currently, it is thought that future attempts at hydrocarbons extraction within Bahamian territorial waters may be politically unfeasible, if not made constitutionally impossible. If, however, hydrocarbons are successfully explored for again, firms may find it easier to earn the trust of local and international conservation groups if the marine environment could be reliably monitored, and active rather than passive methods of oil spill detection and remediation schemes devised [86,87]. An additional example of blue injustice (4, 5, 8, and 9) may be taken from accusations of discrimination due to amendments made to the Fisheries Bill 2020 that explicitly restricts non-Bahamians from participating in commercial fishing within Bahamian waters [54], that affected Bahamian women married to non-Bahamians, and their families.

Noting that Torre-Castro [88] argued that the inclusion of women into SSF management process by consciously and explicitly considering gender and the diversity of actors in coastal and ocean management could lead to better governance and policy reform, solving these specific issues would be excessively complex and are beyond the scope of this paper. For the time being, the recommendations to alleviate or eliminate the injustices given by Bennet et al. [5] are worth repeating verbatim:

1. Recognize and protect resource and spatial tenure and access rights.
2. Take a precautionary approach to reduce pollution and ensure that environmental burdens are not placed on marginalized populations.
3. Minimize the impacts of development on habitats, resources, and ecosystem services.
4. Consider and safeguard the access rights and livelihoods of small-scale fishers.

5. Maintain and promote access to marine resources needed for food security and wellbeing.
6. Develop policies and mechanisms to foster and ensure the equitable distribution of economic benefits.
7. Monitor, mitigate, and manage the social and cultural impacts of ocean development.
8. Recognize, include, and promote the equal role of women in the ocean economy.
9. Recognize and protect human and Indigenous rights.
10. Develop inclusive and participatory planning and governance processes for ocean development.

*3.2. Ocean Observations*

Sustained ocean observations support not only essential scientific research, but also undergirds a diverse range of activities related to safety, operational efficiency, and governance [89–93]. Limited resources and capacity of The Bahamas government to independently conduct and sustain research would restrict the expansion of blue economy activities. This is seen in other jurisdictions, such as Mauritius, where Komul et al. [94] found a limited ability to collect data and effectively analyze it, and a lack of collaborative research and other cost-effective approaches hampered the protection of the nation's SSFs. Other SIDS were characterized as having similar problems where, crucially, using the Mauritian SSFs as an example, they found that SIDS were hampered. These issues were attributed to a lack of qualified personnel, specialized equipment, and funding. Several National Buoy Data Center (NDBC) buoys located in the Atlantic Ocean east of the Lucayan Archipelago (Figure 3) provide in situ observations, but no platforms in the nearshore currently exist, thus requiring other sources of data. For convenience and completeness, land-based Bahamas Department of Meteorology and a C-MAN station owned and maintained by the NDBC that can be used for wind energy analyses are also plotted. Corresponding statistics are given in Table 2. It can be observed that each buoy lies hundreds of kilometers away from the nearest Bahamian island, which renders their observations useless for nearshore applications. Similarly, operational land-based platforms are also limited and concentrated in the northern Bahamian islands where most of the country's population resides. Numerical models or reanalysis data can be used to supplement this information gap, but in each case, require observations to validate their products. These models, however, require significant computational resources and technical expertise to set up, tune, validate, and convert into a format readily digestible for decision-makers, none of which, to the author's best knowledge, currently exists.

**Table 2.** Relevant statistics for National Buoy Data Center (NDBC) buoys and the C-MAN station, and operational Bahamas Department of Meteorology (BDM) automatic weather stations. The distance to land is given as the distance between the buoy and the nearest inhabited Bahamian island.

| ID | Owner | Latitude (°N) | Longitude (°W) | Anemometer Elevation (m) | Water Depth (m) | Distance to Land (km) |
|---|---|---|---|---|---|---|
| SPGF1 | NDBC | 26.704 | 78.995 | 6.6 | - | N/A |
| 41010 | NDBC | 28.878 | 78.485 | 4.1 | 890 | 243 |
| 41047 | NDBC | 27.514 | 71.484 | 4.1 | 5321 | 562 |
| 41043 | NDBC | 21.030 | 64.790 | 3.8 | 5262 | 870 |
| 41046 | NDBC | 23.822 | 68.384 | 3.8 | 5549 | 475 |
| 78059 | BDM | 26.41 | 78.59 | 10 | - | - |
| 78062 | BDM | 26.55 | 78.70 | Unknown | - | - |
| 78072 | BDM | 25.04 | 77.18 | 10 | - | - |
| 78073 | BDM | 25.05 | 77.47 | Unknown | - | - |
| 78075 | BDM | 25.29 | 76.40 | 5 | - | - |
| 78080 | BDM | 24.45 | 76.09 | 10 | - | - |
| 78089 | BDM | 24.04 | 74.31 | 10 | - | - |
| 78091 | BDM | 23.33 | 75.52 | 10 | - | - |

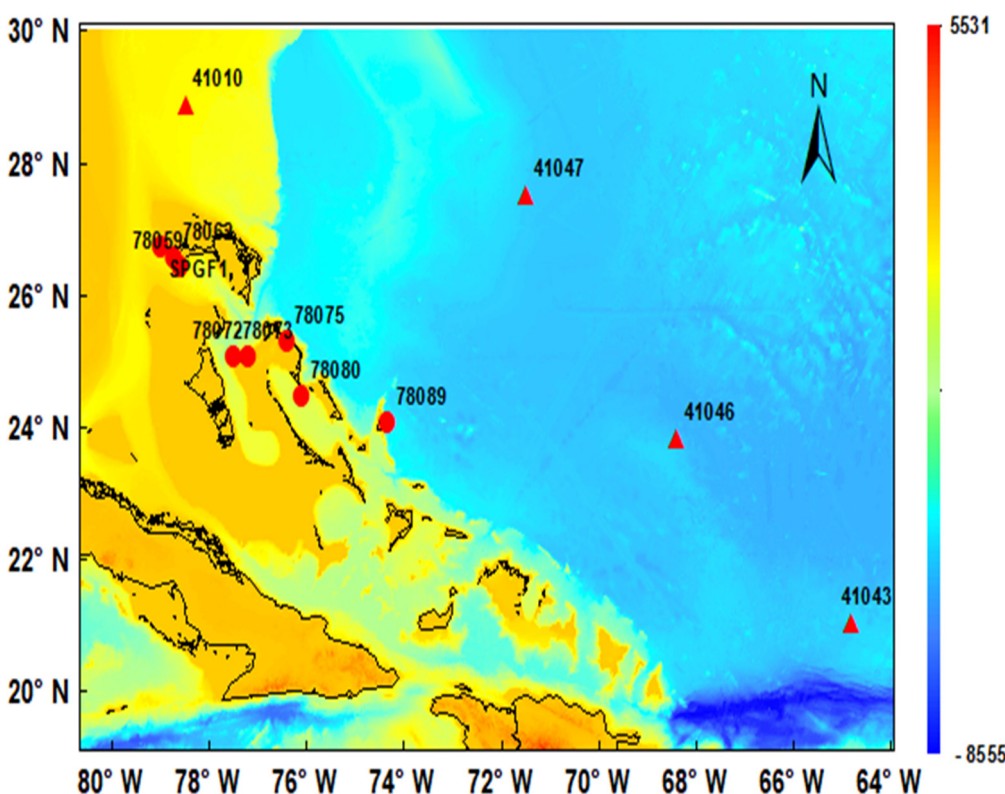

**Figure 3.** Topographic/bathymetric map (shading, units: m) of the Caribbean Sea showing the location of National Buoy Data Center buoys (triangles) and C-MAN and operational Bahamas Department of Meteorology automatic weather stations (circles) in or near The Bahamas.

Government-owned dedicated research vessels, again, to the author's best knowledge, are not in service. Each of the four aforementioned sectors (including others not discussed) can be supported by marine environment monitoring and would thus require large investments into in situ and remote sensing platforms (e.g., automatic weather stations, buoys, and high-frequency/X-band/S-band coastal radar, respectively), and high-performance computing facilities to support numerical (ocean) weather prediction models. The comprehensive usage of the large library of synthetic aperture radar (SAR), amongst other satellite platforms, should also be encouraged to maximize the scope of environmental monitoring at very low costs, deepening the pool of qualified professionals in satellite oceanography. Additionally, for the short term, the network of Kelvin Hughes long-range coastal radars, donated to The Bahamas by the United States Government for maximizing maritime security, as enforced by the Royal Bahamas Defence Force, in the southern Bahamas can be co-opted for oceanographic research. A variety of applications can be derived from X-band (8–12 GHz) and S-band (2–4 GHz) coastal radars. X-band marine radar sensor images of the sea surface are pregnant with information concerning wind speed/direction, wave parameters, and surface currents [95,96], all crucial physical elements of MPAs and ORE resource assessments. S-band marine radar can be used for the same applications but is not hampered by precipitation and severe weather and can be made complementary to X-band systems [97]. High-frequency radars, through slightly different physical principles to X- or S-band marine radar, allow for the same high-resolution oceanographic measurements and find applications in pollution assessments, coastal zone management, tracking environmental change, marine safety, oil spill response, and numerical simulations of three-dimensional circulations, in addition to supporting new understandings of coastal ocean dynamics [98]. Short- and long-term benefits are possible for, and through, the Blue Economy, but are dependent on strategic and tactical investments into ocean observation networks.

## 4. Discussion

Due to the dearth of information, excluding highly specific locations and applications, no specific dollar value can be placed on the economic potential of current and emerging blue economy activities for The Bahamas without irresponsible speculation. It is still possible, however, based off the experience of other jurisdictions, inclusive of their successes, failures, and the lessons they both teach, to know that economic recovery and revitalization is possible with focused attention to the blue economy. This is especially the case given the USD 200 million IDB loan and the economic crashes caused by the landfalling of Hurricane Dorian in late 2019, followed swiftly by the COVD-19 pandemic in early 2020. Indeed, although COVID-19 is often discussed in the context of the negative effects on society, the global shutdown of economic development did have positive effects on both natural and human ecosystems, and it has presented an opportunity for SIDS and coastal governments around the world to retool economies to engender greater pre-COVID-19 growth. For this reason, in addition to the fulfilment of several SDGs (e.g., SDGs 2, 3, 7, 9, 11–14, and 17), the Government of The Bahamas is strongly advised to prepare and approve a Blue Economy Strategic Framework Roadmap and Implementation Plan aimed at outlining a sustainable ocean-based development pathway. The Government is thus encouraged to incentivize private entities to supply capital investments, take a regulatory role in facilitating a favorable ecosystem for renewable energy development, and to foster its ability to manage distributional costs and benefits of future energy investments. Ocean literacy is a major facet of engaging the public that the Government serves and should be made a priority alongside the interdisciplinary challenges that restrict industry development. These include, but naturally are not limited to, the availability of skilled and experienced personnel, technology, relevant policies and regulations, and strategies to assess and mitigate potential environmental and social costs. Increased financial support to fisheries and aquaculture activities will unlock the potential of marine bioprospecting and biotechnology. This has implications for food security, conservation, and drug discovery, but these and other activities (e.g., offshore solar (thermal and photovoltaic), wind, wave, and ocean thermal energy resource assessments) are reliant on in situ and remote ocean marine environment observation platforms. As such, funds funneled into the capitalization of buoys and high-frequency and other types of coastal radars for marine environmental monitoring. Financial plans for their operation and maintenance are put in place to guarantee long-term operation. Multisector stakeholder agreements between the government and its public utilities, private organizations, and environmental protection groups can be drafted to shoulder long-term expenses with the commitment to share data and equipment utilization. Additional economic diversification can occur if private and public corporations, in the form of data assembly centers and information coordination entities, supply direct support for observations. Their activities can be augmented by further data aggregation, integration, and service delivery. In this way, science, operational, policy, and public end-users can all benefit. Investments into the requisite computational resources should be prioritized and schemes, using machine learning and artificial intelligence, be devised to minimize the total computational expense of research activities.

**Author Contributions:** Data curation, B.J.B.; formal analysis, B.J.B. and Y.B.; funding acquisition, D.T.; investigation, Y.B.; methodology, B.J.B.; resources, Y.B.; supervision, D.T.; visualization, B.J.B.; writing—original draft, B.J.B.; writing—review and editing, B.J.B., Y.B., and D.T. All authors have read and agreed to the published version of the manuscript.

**Funding:** This research was funded by The Open Fund of the China Institute of Manufacturing Development, Nanjing University of Information Science and Technology, P.R. China, grant number SK2020-0090-11.

**Institutional Review Board Statement:** Not applicable.

**Informed Consent Statement:** Not applicable.

**Data Availability Statement:** No new data were created or analyzed in this study. Data sharing is not applicable to this article.

**Acknowledgments:** Thanks are extended to Mary Butler of the Bahamas Department of Meteorology for supplying automatic weather station statistics and data. Khaula F. Reid, Marissa Russell, Ian Brown, Abdul Knowles, and three anonymous reviewers are thanked for their insightful comments that greatly improved the quality of this manuscript.

**Conflicts of Interest:** The authors declare no conflict of interest. The funders had no role in the design of the study; in the collection, analyses, or interpretation of data; in the writing of the manuscript, or in the decision to publish the results.

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
