# Peer review of "Blue Economy and Blue Activities: Opportunities, Challenges, and Recommendations for The Bahamas"

_water, doi:10.3390/w13101399_

Round 1

Reviewer 1 Report

I read the manuscript with interest, appreciated the analysis of the state of the art and the mention of Blue Injustice –which I believe is a very actual topic. Some discussion remains quite shallow, and the structure could be strengthened; my recommendations are in this sense:
I think that the potential role of islands as model systems should be stressed and linked to initiatives targeting islands in this respect. For instance, islands need different infrastructure and logistics than mainland states, even in case of same coastal and EEZ extension. Small islands developing states, mentioned in abstract, are the focus for specific actions, and the analysis carried out in the manuscript should tie them to the peculiarity of these systems.
I do also believe that the paper would benefit from a social-ecological approach, i.e. defining the system and its boundaries, and more importantly which are the external and which are the internal drivers. So far, drivers are mentioned but it remains vague (at least for non-local readers like me) the extent to which a change in drivers could bring to a change to the system. The use of shallow and deep leverage points could also be applied in the section regarding final remarks.
The papers by Osterblom et al. https://academic.oup.com/bioscience/article/63/9/735/260704 and https://link.springer.com/content/pdf/10.1007/s10021-016-9998-6.pdf could help to discuss proximate and distal drivers, and perhaps also a figure adapted to the Bahamian situation would be appropriate. I liked the gap analysis and the fact that lack of information was highlighted.
Circularity is rapidly becoming a keyword for sustainable development, and again for island it would make even more sense –was it ever considered in the Bahamian context? Even small examples should be showcased (see e.g. the Ellen McArthur Foundation approach).
In general, the language was quite hard to follow and I would recommend a careful language editing. Also, toponyms and river names were mentioned but there was no indication on the map provided.
The final remarks should go beyond theoretical and provide operational goals, speaking to managers (e.g. targets, KPIs, proposed monitoring) and be in general framed into a strategy.
Comments by paragraph:
Introduction
Please make it broader in scope, linking the Blue Economy to MDGs and mentioning all the paragraphs –not only tourism. The mention of Covid-19 should remain limited to the fact that it disrupted the system somehow, and out of the crisis new opportunities shall emerge.
2.1
LL104-105, not sure about the meaning: is it about supporting domestic tourism and at the same time promoting sustainable tourism? Why should sustainable tourism not be promoted in any context?
Indeed many Countries saw in 2020 a boom of domestic tourism towards seaside locations. Did the survey provided other insights about the reasons of this choice (safety – regulations preventing trips abroad – lower costs – attitude towards promotion of own Country...).
How many Bahamians own touristic activities and how many are employed by foreign tourism companies?
2.2
L120 I cannot understand “the loci of”
What is the rate of offshore vs. Longshore fishery?
Are there lessons to be learned from traditional fishery?
LL158-164 I cannot understand
LL181-186 I cannot understand
2.3
Is it possible to include a mention of FAIR data approach, for data usage and sharing? It is becoming increasingly important in science-to-policy interface, and specially for omics such as in the microB3
2.4
A mention of how wind and wave assessments were made is missing.
It is not strictly blue energy, though a mention of solar thermal and solar photovoltaic should be made.
Please do not forget that the energy demand should also be reduced, not only shifted from fossil fuels. Which are the options for the system to be less energy-demanding? Circularity? Smart mobility?
3.1
LL333-338 I cannot understand
3.2
A discussion based on long vs. short dynamics, and pulse vs. pressure impacts observed should be made: which ones are being targeted? Which ones can be caught by the current system?
Discussion
Reads very detached from the introduction. They should be tied together and bring the reader to a global overview.

Reviewer 2 Report

The authors had an interesting idea for the article. The paper is properly structured and has the correct, up-to-date references. The authors presented problems related to the blue economy and activities for the Bahamas. This is a review article describing the current state opportunities, challenges, and recommendations.
The article is not very scientifically sophisticated, however, as a case study, it brings value to environmental management and is valuable to stakeholders, NGOs, and governmental units.

The topic is original, the paper is a detailed and broad analysis of the study area. However, the paper shows the problem in a local rather than a global scale. 

The article is readable and clear.

I think that the impact of the Covid-19 pandemic on the topic discussed a the article needs to be supplemented. According to many authors and sources, the COVID-19 pandemic has both positive and negative influences on the environment and ecomomy:

Braga, F.; Scarpa, G.M.; Brando, V.E.; Manfè, G.; Zaggia, L. COVID-19 lockdown measures reveal human impact on water transparency in the Venice Lagoon. Sci. Total Environ.2020,736, 139612.

El Zowalaty, M.E.; Young, S.G.; Järhult, J.D. Environmental impact of the COVID-19 pandemic—A lesson for the future. Infect. Ecol. Epidemiol. 2020,10, 1768023.

Dąbrowska, J.; Sobota, M.; Świąder, M.; Borowski, P.; Moryl, A.; Stodolak, R.; Kucharczak, E.; Zięba, Z.; Kazak, J.K. Marine Waste—Sources, Fate, Risks, Challenges and Research Needs. Int. J. Environ. Res. Public Health 2021, 18, 433.

This storyline should be expanded in the paper. 

Reviewer 3 Report

The paper is written on how effective utilization of blue economy and engaging in blue activities may help to revive the economy of Bahamas. I found the paper interesting and more importantly, it explains the future scope of research very well. I recommend to accept the article in it's present form considering it's importance to economic revival in the post-Covid phase. 
